# MicroRNAs Associated with a Bad Prognosis in Acute Myeloid Leukemia and Their Impact on Macrophage Polarization

**DOI:** 10.3390/biomedicines12010121

**Published:** 2024-01-07

**Authors:** Laura Jimbu, Oana Mesaros, Corina Joldes, Alexandra Neaga, Laura Zaharie, Mihnea Zdrenghea

**Affiliations:** 1Department of Hematology, Iuliu Hatieganu University of Medicine and Pharmacy, 8 Babes Str., 400012 Cluj-Napoca, Romania; mesaros.oana@umfcluj.ro (O.M.); a_j_corina@yahoo.com (C.J.); neaga.alexandra@umfcluj.ro (A.N.); zaharie.laura.cristina@elearn.umfcluj.ro (L.Z.); mzdrenghea@umfcluj.ro (M.Z.); 2Department of Hematology, Ion Chiricuta Oncology Institute, 34-36 Republicii Str., 400015 Cluj-Napoca, Romania

**Keywords:** acute myeloid leukemia, macrophage polarization, bad prognosis, microRNA

## Abstract

MicroRNAs (miRNAs) are short, non-coding ribonucleic acids (RNAs) associated with gene expression regulation. Since the discovery of the first miRNA in 1993, thousands of miRNAs have been studied and they have been associated not only with physiological processes, but also with various diseases such as cancer and inflammatory conditions. MiRNAs have proven to be not only significant biomarkers but also an interesting therapeutic target in various diseases, including cancer. In acute myeloid leukemia (AML), miRNAs have been regarded as a welcome addition to the limited therapeutic armamentarium, and there is a vast amount of data on miRNAs and their dysregulation. Macrophages are innate immune cells, present in various tissues involved in both tissue repair and phagocytosis. Based on their polarization, macrophages can be classified into two groups: M1 macrophages with pro-inflammatory functions and M2 macrophages with an anti-inflammatory action. In cancer, M2 macrophages are associated with tumor evasion, metastasis, and a poor outcome. Several miRNAs have been associated with a poor prognosis in AML and with either the M1 or M2 macrophage phenotype. In the present paper, we review miRNAs with a reported negative prognostic significance in cancer with a focus on AML and analyze their potential impact on macrophage polarization.

## 1. Introduction

The central dogma of molecular biology was first presented by Francis Crick in a lecture in 1957, where he stated that deoxyribonucleic acid (DNA) is transcribed into ribonucleic acid (RNA) and RNA is translated into proteins [1]. Of course, at that time, the picture was far less sharp than it is today, but Crick was the first to understand the DNA–RNA–protein sequence. 

The current understanding of nucleic acids recognizes two main categories of RNAs: coding RNAs, represented by messenger RNA (mRNA) and corresponding to the function described above, namely an intermediate template for the translation of genetic information into amino acid sequences, and non-coding RNAs (ncRNA), whose functions have only more recently been partly understood. According to their functions, ncRNAs are further divided into housekeeping ncRNAs and regulatory ncRNAs (Table 1).

MiRNAs are single-stranded short fragments (19–24 nucleotides) of RNA and their main role is the posttranscriptional regulation of gene expression by mRNA silencing; thus, they have many varied physiological roles [5]. Interestingly, one miRNA is able to regulate more than one gene, sometimes even hundreds, due to the fact that they attach with incomplete complementarity to the mRNA [6]. MiRNAs have not only been associated with physiologic processes, but their dysregulation was also associated with cancer and various nonmalignant diseases [7,8,9,10]. 

The biogenesis of miRNA can either be canonic or non-canonic [11]. The canonic pathway starts in the nucleus with the primary transcript (pri-microRNA) that is transcribed by either polymerase II or polymerase III [12]. These precursors are usually found in introns or in intragenic regions. An endonuclease called Drosha, together with the double-stranded RNA-binding protein DiGeorge syndrome critical region gene 8 (DGCR8), will cleave this transcript, resulting in a pre-microRNA [6,12]. Via the exportin 5/RanGTP complex, the pre-microRNA will exit the nucleus and travel to the cytoplasm, where a second endonuclease called Dicer will cleave the loop of the hairpin of the pre-microRNA and a mature microRNA duplex will be formed. Argonaute 2 (Ago 2) will cleave the double-stranded RNA into single-stranded RNA. Later, one of the single strands of RNA will be coupled with the RNA-induced silencing complex (RISC) protein complex (formed by transactivating response RNA-binding protein, Ago2, and Dicer), while the other will be degraded. If the single-stranded RNA is complementary to an mRNA, the mRNA will be degraded via the RISC protein complex by cleavage [12]. There are several non-canonical pathways, which are either Drosha- or Dicer-independent. These pathways also lead to inhibition of translation [11]. 

The history of miRNAs started in the 1980s but peaked in 1993 with the discovery made in Ambros and Ruvnkun’s labs, who described the first microRNAs encoded by *lin-4* [13,14]. Their discovery was made by working with *Caenorhabditis* (*C.*) *elegans*, a type of nematode. In order for *C. Elegans* to progress from stage L1 to L2, the lin-14 protein (p) has to decrease. The regulation of the *lin-14 gene* is performed in a posttranscriptional manner via lin-4, which encodes for two small RNA molecules which will bind, via complementarity, to the *lin-14* gene and will downregulate the production of lin-14, thus allowing the larvae to pass from L1 to L2, a more mature state [13,14]. Later, other miRNAs were described, mounting to presently over a thousand, some present only in one species while others, such as lin-17 [15], are ubiquitous and present in flies to humans [16]. Posttranscriptional gene silencing was also described in plants [17] in 1990, and in 1992, in fungi [18]. 

The main focus of the present paper is to investigate the association of miRNA expression and macrophage polarization in the setting of cancer with focus on AML. In particular, we will focus on miRNAs reported as being associated with a bad prognosis in this group of diseases and how they correlate with macrophage polarization. The goal of this approach is to assess a potential mechanism of miRNA expression affecting the outcome of AML, by either promoting an M2- or an M1-type microenvironment. In a previous paper, we described the miRNAs associated with a good prognosis in AML and their known effects on macrophages [19]. 

## 2. MiRNAs and Cancer

After scientists caught a glimpse of miRNAs in physiologic processes, they started to notice an association between them and different diseases. The first report suggesting the association between miRNAs and cancer, specifically chronic lymphatic leukemia (CLL), was published in 2002 by Croce and colleagues [8]. In their paper, they demonstrated that microRNA (miR)-15 and miR-16 are downregulated in CLL. Both miRNAs were found on chromosome 13q14, a region frequently deleted in several other cancers [20,21,22,23]. Apparently, more than 50% of the miRNAs are situated in ‘fragile sites’ of the genome that are associated with malignancies. The miRNAs that were associated with the development and progression of cancer were named ‘oncomiRs’ [24]. Of course, miRNAs are not only associated with cancer but also with protecting the host against cancer, and thus acting as tumor-suppressors [24]. After Croce’s discovery, a substantial amount of information on miRNAs and cancer emerged. 

Based on the information that miRNAs have the ability to act both as an oncomiR and as a tumor suppressor, clinical trials targeting them have been developed [25]. A quick search on www.clinicaltrials.gov (accessed on 2 January 2024) using ‘cancer’ and ‘miR’ showed more than 400 results, mostly focused on miRNAs as biomarkers. 

AML is a hematological malignancy with a dismal prognosis. Many reports presented the association between miRNA expression and specific chromosomal abnormalities [26,27] or mutations in AML [28,29,30]. On the other hand, miRNAs were also associated with prognosis, disease progression, or proliferation [31,32,33,34]. Based on these findings, miRNAs are now regarded as important biomarkers that might predict the response to treatment, the risk of progression, or the rate of complete remission (CR) and overall survival (OS) in both solid cancers of hematological malignancies [35,36,37,38]. Only a few studies have focused on the role and association between miRNAs and macrophage polarization in cancer [19,39,40,41]. 

In AML, several miRNA targets have already been addressed. One of them, miR-126, is highly expressed in patients with inv(16)(p13q22). In a mouse model of inv(16)(p13q22) AML, Kuo et al. showed that adding an miR-126 inhibitor improved survival compared to the control group. Another study showed that adding an miR-10a inhibitor to cytarabine + nutlin-3a, a murine double minute 2 homolog (MDM2) inhibitor, improved OS [42]. MiR-193b is a tumor suppressor in AML, inhibiting excessive proliferation. In AML, miR-193b is downregulated. An miR-193b mimic showed promising results in AML xenografts compared to controls, improving survival [43]. Also downregulated in AML is miR-146 [44]. A study published in 2020 reported that an miR-146 mimic successfully targeted the NF-κB pathway in mice by inhibiting it and also inhibiting TRAF6 and IRAK1, thus inhibiting progression. The same study showed that the miR-146 mimic inhibited proliferation in different cell lines [45]. Another targeted miRNA was miR-9, which is overexpressed in mixed lineage leukemia (MLL). Adding an miR-9 inhibitor in the Mono-Mac-6 cell line has been proven efficacious, decreasing proliferation [46].

## 3. Macrophages and Their Polarization

Macrophages are part of the innate immunity, playing an important role in phagocytosis, thus eradicating infections and tumor cells, and also in tissue repair [47]. A simplistic yet commonly used classification divides macrophages into two categories based on their polarization: M1 macrophages, classically activated with a pro-inflammatory activity, and M2 macrophages, alternatively activated and anti-inflammatory [48]. Polarization refers to the ability of macrophages to exhibit different functions and phenotypes based on stimuli from the microenvironment. Cytokines such as lipopolysaccharide, interferon (IFN)-γ, and tumor necrosis factor (TNF)-α will tilt the neutral, non-polarized M0 phenotype towards M1 polarization, while apoptotic cells, IL-4, IL-13, IL-10, IL-33, fungi, parasites, and transforming growth factor (TGF)-β will promote M2 macrophages [49,50]. Based on their phenotypes, M1 macrophages express on surface markers such as CD64 and CD86, while M2 macrophages express CD206 and CD163 [51]. The imbalance between M1 and M2 macrophages can either lead to inflammatory states or to cancer due to a very ‘tolerant’ state [51,52]. In cancer, macrophages are very abundant in the tumor environment (TME) and they are frequently called tumor-associated macrophages (TAMs) [53]. In early stages, M1/M2 ratios favor the M1 subtype, but in later stages, the M2 subtype is more common and is associated with a poorer prognosis [54]. A high M2/M1 ratio is associated with proliferation, tumor evasion, angioneogenesis, metastasis, immune suppression, and, overall, a poor prognosis [51].

## 4. MicroRNAs Associated with a Bad Prognosis of AML and M1 Polarization

The miR-17-92 cluster (miR-17-5p, miR-17-3p, miR-18a, miR-19a, miR-19b, miR-20a, and miR-92-1), or oncomiR-1, is highly dysregulated in several solid and hematological cancers [33]. This cluster regulates a plethora of transcription factors such as TP53, c-MYC, n-MYC, STAT3, MXI1, E2F1, E2F2, E2F3, and NKX3.1. Furthermore, the miR-17-92 cluster regulates TGF-β receptor II, Smad2 and Smad4, BCL2L11, anti-angiogenic factors thrombospondin-1 (TSP-1), connective tissue growth factor (CTGF), insulin gene enhancer protein (Isl-1), and the T-box 1 protein (Tbx1) [55]. In AML, the miR-17-92 cluster is overexpressed and is associated with a poor prognosis. MiR-92a-1-5p was increased in mouse bone-marrow-derived M1 macrophages [56]. On the other hand, downregulation of miR-17-92 is essential for myeloid differentiation [57]. Thus, in MLL, where the abovementioned miRNAs are overexpressed, targeting both miR-17-5p and miR-19a-3p by antagomiRs reduces the ability of MLL cells to form colonies compared to non-MLL AML controls [58]. Several other studies suggested that the miR-17 cluster is associated with a poor prognosis [33,59] through several mechanisms, including the promotion of leukemic blast proliferation [60]. With regard to macrophage polarization, miR-17 and miR-20a promote an M1 phenotype by inhibiting signal-regulatory protein a (SIRPa) [60]. 

MiR-20a is part of the miR-17-92 cluster and is associated with several cancers. It has been reported that miR-20a interferes with signaling pathways such as ENH1/Id1, MAPK1/c-Myc, PTEN/PI3K/AKT, FBXL5/BTG3, or the Sonic hedgehog pathways [61]. In cancers, the overexpression of miR-20a has been associated both with a poor [62,63] or a good prognosis [64]. Also, its downregulation was associated with a poor prognosis [65]. In a cohort of 61 patients with AML, downregulated miR-20a was associated with a poorer prognosis, compared to those with a high expression [66]. In cell lines, overexpression of miR-20a stimulated apoptosis and inhibited proliferation of AML cells [66], thus suggesting its potential as a therapeutic target. MiR-20a promotes M1 macrophage polarization by inhibiting SIRPa [60,67].

MiR-125b is associated with apoptosis, differentiation, and proliferation by regulating important pathways such as NF-κB, p53, PI3K/Akt/mTOR, ErbB2, and Wnt [68]. Interestingly, miR-125b also targets DICER1, suggesting that its alteration could affect the entire biogenesis of miRNAs. Similar to other miRNAs, miR-125b behaves either as an oncogene or as a tumor suppressor, being either upregulated in certain cancers or downregulated in others. In both AML and acute lymphoblastic leukemia (ALL), miR-125b is generally upregulated [68]. In AML and myelodysplastic syndrome (MDS) patients with t(2;11)(p21;q23) translocation, miR-125b was 90 times more upregulated compared to normal controls. Several subtypes of AML, like *fms-related receptor tyrosine kinase 3* (*FLT3*)-mutated AML, AML harboring the translocation AML-ETO, acute promyelocytic leukemia, and trisomy 21-acute megakaryocytic leukemia, are associated with high levels of miR-125b [69]. Another study showed that miR-125b is highly implicated in myeloid differentiation and erythroid and megakaryocytic progenitor proliferation. In acute megakaryocytic leukemia associated with Down’s syndrome, myeloid differentiation was severely impaired [70]. Interestingly, miR-125b is highly expressed on macrophages, and the studies performed by Chaudhuri et al. showed that overexpression of miR-125b in mice injected with EL4-Fluc thymoma cells increased the macrophages’ capacity for killing and led to tumor shrinkage. MiR-125b regulates macrophage activation via IRF4 [71]. There are conflicting results about miR-125b. Another study showed that mice which overexpressed miR-125b in transplanted liver cells developed various hematological malignancies such as B-ALL, T-ALL, and myeloid neoplasms [72]. In mice, overexpression of miR-125b was associated with the development of myeloproliferative neoplasms and with transformation to AML [73]. In AML, miR-125b is associated with refractoriness to daunorubicin and, thus, with a poorer prognosis, altering apoptosis via decreasing PUMA and GRK2 [74]. Concerning macrophage polarization, miR-125b is associated with the M1 phenotype [75].

MiR-146a and b are situated on chromosomes 5 and 10, respectively, and they are dysregulated in several cancers [76], promoting proliferation and metastasis [77]. A study in miR-146b-knockout mice showed that they developed AML or B-cell lymphoma, probably due to the alteration of the NF-kB pathway by inhibiting TNF receptor-associated factor 6 (TRAF6) and interleukin-1 receptor-associated kinase 1 (IRAK1) [78]. In AML, downregulation of miR-146a was associated with progression by targeting the NF-κB pathway [79]. On the other hand, a study of 53 AML patients showed that miR-146a is overexpressed and is associated with a poorer prognosis [80]. In pediatric AML with a normal karyotype, hsa-miR-146b was associated with a poor prognosis [81]. Lower levels of miR-146b were detected in low- and intermediate-I-risk MDS compared to intermediate-II- and high-risk MDS [82]. MiR-146b targets IRF5, a transcription factor, promoting the M1 macrophage phenotype [83]. 

MiR-155 is encoded by a sequence located on the 21st chromosome in the non-coding *B cell integration cluster (BIC) gene* [84], which is heavily dysregulated in different types of cancers. MiR-155 is also involved in regulating both innate and adaptive immune responses, having an important role in myeloid progenitor differentiation by targeting the transcription factor PU.1. Apart from PU.1, miR-155 also regulates SHIP1, which acts as a negative regulator of the PI3K/Akt pathway involved in several biological processes such as differentiation, apoptosis, transcription, and translation [85]. In lung cancer, a meta-analysis showed that miR-155 could be useful in diagnosis, but could not predict the response to treatment [86]. On the other hand, high levels of miR-155 in breast cancer were associated with a good prognosis and with a good response to immunotherapy [87]. In patients with diffuse large B-cell lymphoma, cobomarsen, an miR-155 inhibitor, showed promising results in preclinical studies by reducing the tumor burden and stimulating apoptosis [88]. Also, in B-cell lymphoma, melanoma, and breast, gastric, ovarian, colon, nasopharyngeal, and pancreatic cancer, overexpression of miR-155 was associated with a better prognosis, acting as a tumor suppressor by unleashing the immune system against the tumor [89]. At the other end of the spectrum, in AML, several studies have proven that upregulated miR-155 is associated with a poor prognosis [90,91]. A study which included 363 patients with AML showed that patients with low levels of miR-155 had a better OS than normal karyotype patients [92]. Similar results have been seen in an AML pediatric cohort of 196 patients with a normal karyotype [93]. From a subtype point of view, high levels of miR-155 were associated with *FLT3-internal tandem duplication (ITD)* positive AML [94], and MLL [95]. Another inhibitor of miR-155, MLN4924, has been used in the AML setting, improving survival in mice. Its mechanism is based on the upregulation of SHIP1, a phosphatase highly expressed on hematopoietic cells which inhibits survival and proliferation [96]. Furthermore, miR-155 upregulates PU.1, a transcription factor which promotes differentiation [97]. Another study showed that silvestrol, a compound extracted from a plant called *Aglaia foveolata*, improved survival in mice with *FLT3-ITD*-mutated AML, decreasing both miR-155 and the expression of *FLT3-ITD* [98]. In addition, silvestrol showed promising results in combination with cytarabine, daunorubicine, and etoposide [99]. MiR-155 also regulates aerobic glycolysis. In cell lines with knockout miR-155, it has been observed that the treatment sensitivity increased for both FLT3 inhibitors and adriamycin by inhibiting aerobic glycolysis via *PIK3R1*, a gene also associated with insulin resistance [100]. In cancer, overexpression of miR-155 was associated with the M1 macrophage subtype [84].

Another miRNA associated with a dismal prognosis is miR-210. Higher levels of miR-210 were associated with a low OS [101]. As with other miRNAs, miR-210 is associated with cell proliferation, angioneogenesis, and DNA repair, and its overexpression is associated with a poor prognosis not only in AML, but also in different solid cancers. Interestingly, in ALL, patients with low levels of miR-210 were associated with relapse and with a lower response to treatment [102]. Also, in MDS, miR-210 and miR-155 downregulate SHIP1, and thus tyrosine-protein kinase Tec is upregulated, promoting MDS cell survival [103]. MiR-210 switches macrophages to M1 polarization [104].

## 5. MicroRNAs Associated with a Bad Prognosis in AML and M2 Polarization

MiR-19a dysregulation has been reported in several cancers [105] and is associated with the upregulation of the NF-κB signaling pathway [106]. MiR-19a was found to be expressed in many human cancers with contradictory consequences, being reported to both promote or inhibit cancer progression in different type of neoplasms [107]. A study by Zhang et al. showed that miR-19a/b was upregulated in AML patients compared to controls. Moreover, overexpression of miR-19a/b was associated with the female gender, elderly patients, several mutations such as *U2AF1*, *C-KIT*, *CEBPA*, and *IDH1/2*, and a poor prognosis (lower CR rates and OS) [107]. MiR-19a inhibits the M1 subtype by targeting *STAT1* and interferon regulatory factor 1 (IRF1) [108] and promotes the M2 macrophage phenotype by activating *STAT3* [109].

MiR-21 has regulatory roles in several biological processes and is expressed in all types of cells. MiR-21 is situated on chromosome 17 [110]. This miRNA is dysregulated not only in cancer, but also in several nonmalignant conditions including cardiovascular and pulmonary diseases [111] and autoimmune conditions [112,113]. Its expression is ubiquitous but with varying levels, being highly expressed in immune cells such as dendritic cell, monocytes, and macrophages [110]. Due to its abundant expression in different cells, miR-21 is not really suitable as a biomarker in any disease. 

In AML, miR-21 downregulates several genes involved in apoptosis, such as *programmed cell death 4 (PDCD4)*, *BTG2*, *SPRY1*, and *PTEN* [114,115]. Another study showed that miR-21 was overexpressed in AML, while Krüppel-like factor 5 (KLF5) was downregulated. KLF5 is a transcription factor which acts as a tumor suppressor in AML. In *nucleophosmin-1 (NPM-1)*-mutated AML, miR-21 levels were higher compared to wild-type *NPM1*, suggesting its role in the diseases pathogenesis [114]. Overexpression of miR-21 was associated with a poorer prognosis [116] and chemo-resistance [115]. Another study showed that in homeobox (HOX)-associated AML, targeting miR-21 and miR-196b improved the prognosis and response to treatment [117]. MiR-21 was shown to promote an M2 macrophage phenotype in different settings either in cancer or in sepsis [118,119]. 

MiR-23a is part of the miR-23a–27a–24-2 cluster and it is encoded on chromosome 19. MiR-23a is not only associated with cancer but also with other conditions such as cardiac [120] or autoimmune diseases, playing an important role in apoptosis, proliferation, and differentiation [121]. With the exception of erythroid leukemia, where miR-23a is upregulated, in other hematological malignancies such as chronic myeloid leukemia or CLL and in other subtypes of AML, miR-23a is downregulated. The same is true in solid cancers, where in most subtypes, miR-23a is downregulated, while in some cancers like head and neck cancer, it is upregulated [120]. Moreover, miR-23a was associated in several studies with advanced stages, metastasis, and a dismal prognosis or resistance to treatment [120]. Interestingly, the miR-23a-27a-24-2 cluster also regulates macrophage polarization. MiR-23a stimulates M1 macrophage polarization in different settings [122,123], while miR-27a and miR-24-2 promote M2 polarization [122]. MiR-27a is also encoded on chromosome 19 and it is associated with apoptosis, proliferation, differentiation, metastasis, angioneogenesis, and treatment response, playing the role of both a tumor suppressor and an oncogene in different types of cancer [124]. In most cancer patients, miR-27a is overexpressed.

In AML, overexpression of miR-23a is associated with chemo-resistance to cytarabine, lowering the expression of *TOP2B*, a gene that encodes a DNA topoisomerase involved in different genetic processes [125]. A high miR-24 expression was reported in AML patients with t(8;21), but with no impact on OS and relapse-free survival (RFS) compared to those with a low miR-24 expression [126]. A study which included 147 patients with acute leukemia demonstrated that miR-24 was overexpressed in both AML and ALL compared to healthy controls and was associated with a dismal prognosis [127]. On the other hand, another study showed that miR-24 has a higher expression on AML cells compared to ALL cells [128].

MiR-221/miR-222 are encoded on the Xp11.3 chromosome and they are highly overexpressed in certain types of cancers, such as glioma, bladder, pancreatic, gastric, or colorectal cancer, and in some hematological malignancies, such as CLL, ALL, multiple myeloma (MM), or AML [129,130]. They are associated with tumorigenesis, angiogenesis, metastasis, and a worse prognosis or chemo-resistance in certain types of cancers [129]. In AML, miR-221 and miR-222 are overexpressed and downregulate *TP53* via YOD1, a deubiquitinase. A lower level of YOD1 is associated with downregulation of *TP53*. MiR-221 and miR-222 are inversely correlated with YOD1 [131]. Targeting miR-221 has been shown to be beneficial in a study where an anti-miR-221 and gold nanoparticles co-carrying AS1411, an actamer, inhibited leukemic growth by targeting the NCL/miR-221/NFκB/DNMT1 pathway [132]. In another study, miR-222 and miR-181 were studied in the AML setting. MiR-222 was highly downregulated but was not associated with the response to treatment or FAB classification [133]. On the contrary, other studies suggest that miR-222 is overexpressed in AML and upregulates the Wnt/β-catenin pathway by inhibiting Axin2, a tumor suppressor [134]. Also, miR-222 and miR-181 had a higher expression in AML than in MDS both in peripheral blood and bone marrow [135]. 

In ovarian cancer, miR-222 induces M2 macrophage proliferation [136]. The same relationship was reported in a study in mice with burn injuries [137].

MiR-126 has been associated with a poor prognosis, treatment refractoriness, and chemo-resistance, targeting the PI3K/AKT/MTOR pathway and thus stimulating proliferation of leukemic cells. In humans, under normal conditions, miR-126 is involved in maintaining the hematopoietic stem cells in quiescence, while in leukemia, the malignant cells are similarly kept in a dormant state. In AML, this miRNA is overexpressed [138]. As expected, inhibiting miR-126 in acute leukemia improved survival by eliminating the malignant cells, while in normal bone marrow, its inhibition stimulated the proliferation of hematopoietic stem cells [139]. Similar results were published by using an antagomiR-126 in AML [140]. High levels of miR-126 were also encountered in the FAB M4Eo AML subtype, and promising results have been shown when using an miR-126 inhibitor [141]. Interestingly, a study from 2015 showed that both overexpression and knockout of miR-126 leads to leukemia by affecting different signaling pathways [142]. Based on these data, miR-126 could be a valuable target in the treatment of AML. In different settings, miR-126 has been associated with the M2 phenotype [143,144,145], but information in cancer settings is lacking.

Figure 1 summarizes the association of all the aforementioned miRNAs with macrophage polarization.

Table 2 presents the miRNAs and their targets (pathways or transcription factors).

## 6. MicroRNAs Associated with a Bad Prognosis in AML and Unknown Macrophage Polarization

MiR-3151 is encoded in the *BAALC gene* in intron 1. A study which included 179 patients with de novo, cytogenetically normal AML of ≥60 years old showed that miR-3151 is overexpressed and is associated with mutations such as *RUNX1* and *MN1*, with wild-type *NPM1*, and with a high expression of the *BAALC gene*. Even if it was associated with a lower percentage of peripheral blast cells, the prognosis was poorer in this category of patients [146]. In younger AML patients, with intermediate-risk AML, similar results have been reported, suggesting that miR-3151 is a potential biomarker and target in AML [147]. Even after allogeneic hematopoietic stem cell transplantation, overexpression of miR-3151 or the *BAALC gene* was associated with a poorer prognosis [148]. No convincing evidence of its association with macrophage polarization exists.

MiR-4262 is also associated with a poor prognosis. Patients with a high expression of miR-4262 had a lower OS and lower relapse-free survival compared to low expressors [149]. MiR-4262 targets KLF6, which in the AML setting is downregulated and thus stimulates proliferation and invasion [150]. We have not found any association between macrophage polarization and the abovementioned miRNA.

## 7. Discussion

AML is a hematological malignancy with a poor prognosis, despite major improvements having been made not only in the management but also in the understanding of the genetic landscape of this disease. Starting in the 1990s, an explosion of data regarding miRNAs emerged, mostly in research proposing their use as biomarkers, both in diagnosis and in the assessment of treatment efficacy. In acute leukemias, as with several other hematologic malignancies, the obtention of diagnostic material is relatively straightforward because the tumor is present in a circulating form and thus can be obtained by drawing blood or by bone marrow aspiration. In contrast, in solid tumors, where the tumor material is sometimes far less easily accessible, circulating plasma miRNAs and liquid biopsies are regarded as promising tools for diagnosis and prognosis. From a therapeutic viewpoint, in preclinical studies, either miRNA inhibitors or miRNA mimics have shown promising results. However, very few have been translated into the clinic. In AML, there are several miRNAs associated with a bad prognosis, most of them upregulated, and thus their inhibition holds promise to improve disease outcomes. On the other hand, macrophages and their M1 or M2 polarization have been associated with the prognosis of both solid cancers and hematological malignancies. In most cases, M2 macrophages are associated with a dismal prognosis, with advanced disease and metastasis. Their involvement in the TME is highly important and thus targeting them could further improve survival. We found that miRNAs associated with a dismal prognosis in AML are not mainly associated with an M2 phenotype, which, in AML too, is linked to progression and a low OS. Thus, miRNAs are likely to influence the prognosis through a number of mechanisms, and not mainly via macrophage polarization.

## Figures and Tables

**Figure 1 biomedicines-12-00121-f001:**
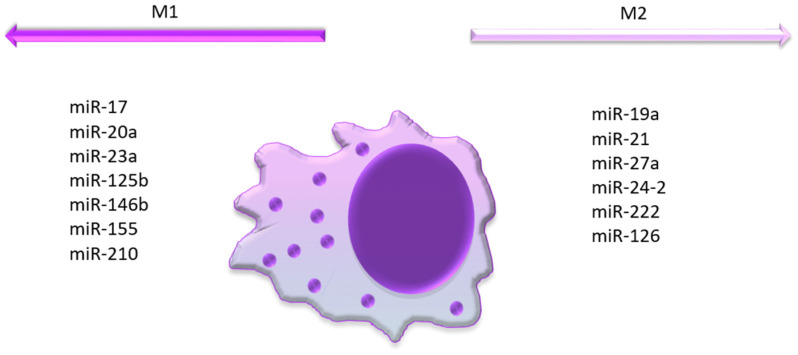
MiRNAs associated with M1 or M2 polarization. MiR-17, 20a, 23a, 125b, 146b, 155, 210 are associated with M1 macrophage polarization, while miR-19a, 21, 27a, 24-2, 222, 126 are associated with the M2 phenotype.

**Table 1 biomedicines-12-00121-t001:** Types of non-coding RNAs. Adapted from [2,3] and from [4] with permission.

	Name	Abbreviation
housekeeping ncRNAs	ribosomal RNA	rRNA
	transfer RNA	tRNA
	small nuclear RNA	snRNA
	small nucleolar RNA	snoRNA
	telomerase RNA	TERC
	tRNA halves	tiRNA
	tRNA-derived fragments	t-RF
regulatory ncRNAs	microRNA	miRNA
	small interfering RNA	siRNA
	piwi-interfering RNA	piRNA
	enhancer RNA	eRNA
	long non-coding RNAs	lncRNA
	circular RNA	circRNA
	YRNA	YRNA

**Table 2 biomedicines-12-00121-t002:** miRNAs and their targets.

miRNA	Pathway/Transcription Factor Targeted
miR-17-92 cluster	*TP53*, c-MYC, n-MYC, *STAT3*, MXI1, E2F1, E2F2, E2F3, *NKX3.1*, TGF-β receptor II, Smad2, Smad4, BCL2L11, TSP-1, CTGF, Isl-1 and Tbx1
miR-20a	ENH1/Id1, MAPK/c-Myc, PTEN/PI3K/AKT, FBXL5/BTG3, the Sonic hedgehog pathways, and SIRPa
miR-125b	NF-κB, p53, PI3K/Akt/mTOR, ErbB2, Wnt, DICER1, IRF4, PUMA and GRK2
miR-146	NF-kB pathway, TRAF6, IRAK1, IRF5
miR-155	PI3K/Akt, PU.1, SHIP1
miR-210	SHIP1
miR-19a	NF-κB, *STAT1*, IRF1, *STAT3*
miR-21	*PDCD4*, *BTG2*, KLF5
miR-23a	*TOP2B*
miR-221	*TP53*
miR-222	Wnt/β-catenin pathway, Axin2,
miR-126	PI3K/AKT/MTOR
miR-4262	KLF6

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
