# Peer review of "MicroRNAs Associated with a Bad Prognosis in Acute Myeloid Leukemia and Their Impact on Macrophage Polarization"

_biomedicines, 2024, doi:10.3390/biomedicines12010121_

Round 1

Reviewer 1 Report

Comments and Suggestions for Authors

This manuscript by Jimbu et al. is a summary of the current knowledge on AML- and macrophage polarization-associated miRNAs. While reading it, I initially found it very difficult to find a connection between the two topics. What does AML have to do with macrophage polarization? Here is clearly missing a description whether M1 or M2 polarization has an inducing or proliferative effect on AML and why the authors want to look for similarities here. Moreover, the following chapters do not present a joint consideration of these two phenomena, but first the influence of the respective miRNA on cancer and then a sentence on polarization. 

There are some inaccuracies in the introduction to miRNAs that definitely need to be corrected (criticisms 1-4).

Major points:

1. line 57, There is also definitely evidence for PolIII transcribed miRNAs. The statement that all miRNAs are transcribed from PolII is too sweeping.

2. line 58, According to the canonical miRNA pathway, the primary miRNA transcript already has a hairpin structure that is recognized by Drosha and Pasha to cleave the pre-miRNA. Here should not be described inaccurately.

3. line 64, The RISC complex is not only the transporter but also the protein moiety that compares the miRNA sequence with the mRNA sequence and performs either silencing or translational inhibition in case of sequence complementarity. 

4. lines 56-80 This whole chapter needs to be corrected. What is written here is not quite correct. Perhaps also add the first papers that described the miRNA phenomenon, such as Romano & Mancini 1992 and Napoli et al 1990.

5. The description of each miRNA is a database-like enumeration of the respective miRNAs followed by the facts known about them. Also, here it does not refer exclusively to AML, but to cancer in general, so it does not do justice to the title. In the conclusion of each paragraph about a miRNA there is then a completely detached description of its influence on macrophage polarization. All this is not really an easy-to-read text. Some examples are given here:

5a: line 178, tumor shrinkage in AML??

5b: line 189, B-cell lymphoma

5c: line 203-207 breast cancer, B-cell lymphoma, melanoma, gastric, ovarian, colon, nasopharyngeal carcinoma and pancreatic cancer

5d: line 230 angiogenesis

5e: line 271 CML

5f: line 289 ALL

5g: line 296 certain types of cancers

6. It would be desirable here to sort the miRNAs according to the target mRNA, so that not all miRNAs are enumerated individually, but these miRNAs are also sorted according to their manipulated pathways. The structure chosen here is more reminiscent of an encyclopedia than a review article and is thus very tedious to read. 

Comments on the Quality of English Language

I suggest the authors of this manuscript to find a native speaker who is a scientifically educated person to revise this manuscript. There are some inadequacies to be found. 

Reviewer 2 Report

Comments and Suggestions for Authors

This manuscript focuses on reviewing the literature regarding the impact of miRNAs on macrophage polarization in AML. Overall, this is a well-written and detailed manuscript. Most parts of the manuscript are well-structured. However, there are some things to be improved:

1) The authors present mRNA biogenesis and its historical background in the 'Introduction'. The authors could also provide a general overview of the mechanisms of action of miRNAs.

2) I would also like to ask you to add some tables. Some summary tables would be useful. The table should contain the target/pathways and function of miRNAs that are associated with bad prognosis in AML.

Round 2

Reviewer 1 Report

Comments and Suggestions for Authors

This manuscript has been significantly improved and presents an overview of the role of miRNAs in cancer with a special focus on AML. I consider this manuscript publishable as is.